# Identification of Adipocytokine Pathway-Related Genes in Epilepsy and Its Effect on the Peripheral Immune Landscape

**DOI:** 10.3390/brainsci12091156

**Published:** 2022-08-30

**Authors:** Jinkun Chen, Ruohan Sun, Di Jin, Quan Yang, He Yang, Yu Long, Lixian Li

**Affiliations:** 1Department of Neurosurgery, The First Affiliated Hospital of Harbin Medical University, Harbin 150001, China; 2Department of Neurology, The First Affiliated Hospital of Harbin Medical University, Harbin 150001, China

**Keywords:** epilepsy, adipocytokine pathway-related genes, random forest, nomogram, consensus clustering, peripheral immune infiltration

## Abstract

Epilepsy is a multifactorial neurological disorder with recurrent epileptic seizures. Current research stresses both inflammatory and autoimmune conditions as enablers in the pathophysiological process of epilepsy. In view of the growing concern about the role of adipocytokines in antiepileptic and modulating immune responses, we aimed to investigate the relevance of the adipocytokine signaling pathway in the pathological process of epilepsy and its impacts on peripheral immune characteristics. In this study, expression profiles of 142 peripheral blood samples were downloaded from the Gene Expression Omnibus (GEO) database. Adipocytokine pathway-related genes were screened out by feature selection using machine-learning algorithms. A nomogram was then constructed and estimated for the efficacy of diagnosis. Cluster analysis was employed for the recognization of two distinct epilepsy subtypes, followed by an estimation of the immune cell infiltration levels using single-sample gene-set enrichment analysis (ssGSEA). The biological characteristics were analyzed by functional enrichment analysis. The aberrant regulation of adipocytokine signaling pathway was found in the peripheral blood of patients with epilepsy. Twenty-one differently expressed adipocytokine pathway-related genes were identified and five (RELA, PRKAB1, TNFRSF1A, CAMKK2, and CPT1B) were selected to construct a nomogram. Subsequent validations of its forecasting ability revealed that this model has satisfactory predictive value. The immune cell infiltration degrees, such as those of innate immune cells and lymphocytes, were found to significantly correlate to the levels of adipocytokine pathway-related genes. Additionally, 239 differentially expressed genes (DEGs) were identified and their biological functions were mainly enriched in the regulation of the immune response. In conclusion, our results confirmed the predictive value of adipocytokine pathway-related genes for epilepsy and explored their effects on immune infiltration, thereby improving our understanding of the pathogenesis of epilepsy and providing assistance in the diagnosis and treatment of epilepsy.

## 1. Introduction

Epilepsy is a chronic neurological disease with obvious heterogeneity, in which the etiology, risk factors, and manifestations diverge significantly [1]. It is characterized by unprovoked epileptic seizures due to transient abnormal neuronal activity [2]. An epidemiologic study shows that there are approximately 50 million people worldwide affected by this disease, and the prevalence seems to have a tendency to increase [3]. Nowadays, an evolving understanding of aetiological and genetic factors, as well as advances in clinical testing methods, has contributed to refining the diagnosis. However, in terms of pharmacotherapy, despite the presence of more than two dozen antiepileptic drugs approved by the U.S. Food and Drug Administration (FDA), about one-third of patients do not respond to these drugs [4]. Thus, further investigation is urgently needed to increase the effectiveness of antiepileptic drug therapy.

Previous studies have placed great emphasis on the role of neuroinflammation in epilepsy. Inflammatory conditions in most autoimmune diseases, such as systemic lupus erythematosus, type 1 diabetes mellitus, myasthenia gravis, and multiple sclerosis, have been found to facilitate the manifestation and risk of epilepsy [5,6]. In turn, epileptic seizures exacerbate the overactivation of innate and adaptive immunity, revealing a bidirectionally interacting network in the pathological process of epilepsy [7].

Adipose tissue is defined as an important endocrine organ involved in energy homeostasis, inflammation, and immunity [8]. Adipocytokines such as leptin, adiponectin, and resistin, as well as cytokines and chemokines, such as tumour-necrosis factor (TNF), interleukin-6 (IL-6), IL-1, and CC-chemokine ligand 2 (CCL2) produced by adipose tissue, have been shown to play key roles in the activation of innate and adaptive immunity [9,10]. A previous study reported reduced plasma leptin levels and increased adiponectin levels within 24 h in patients who suffered from seizures [11]. Xu et al. found that both intranasal leptin delivery and direct injection of leptin into the cerebral cortex could result in partial excitatory inhibition of hypothalamic neurons, thus exerting an antiepileptic effect [12]. Besides, the antiepileptic effect of the ketogenic diet (KD) in drug-resistant epilepsy (DRE) also appears to be associated with adipocytokine levels [13].

The application of several adipocytokines in epilepsy has been preliminarily explored in recent years. However, more work needs to be done to fully comprehend the relationship between epilepsy and adipocytokines. In this study, we downloaded expression profiling data from the Gene Expression Omnibus (GEO) database and systematically assessed the activation of the adipocytokine signal pathway in the peripheral blood of epileptic patients. Through a series of machine learning and bioinformatics analysis methods, we attempted to identify adipocytokine pathway-related genes as diagnostic biomarkers for epilepsy. Moreover, we investigated the immune cell infiltration under specific gene expression patterns and further discussed its biological characteristics.

## 2. Materials and Methods

### 2.1. Data Preparation

The gene expression profiles of 142 peripheral blood samples, including 91 epilepsy samples and 51 normal samples, were downloaded from the Gene Expression Omnibus database and the serial number was GSE143272 (https://www.ncbi.nlm.nih.gov/geo/query/acc.cgi?acc=GSE143272; accessed on 20 March 2022). The platform was GPL10558 Illumina HumanHT-12 V4.0 Expression BeadChip. Gene probes were converted into genetic symbols. Data were preprocessed using the “limma” [14] package in R (version 4.1.2). Genes with duplicative expression values were calculated as the median, and genes without matching expression values were excluded.

### 2.2. Gene Set Enrichment Analysis

In our study, the Gene Set Enrichment Analysis was performed using GSEA software (v4.2.3) downloaded from the official website (https://www.gsea-msigdb.org/gsea/index.jsp; accessed on 20 March 2022). The subset of C2 (c2.cp.kegg.v7.5.1.symbols) achieved from the Molecular Signatures Database was chosen for running GSEA between normal and epilepsy samples.

### 2.3. Difference Analysis of Adipocytokine Pathway-Related Genes

A gene list of adipocytokine signaling pathways was acquired from the Kyoto Encyclopedia of Genes and Genomes (KEGG) pathway database (hsa04920). A total of 43 genes were detected in the peripheral blood samples. We compared the expression differences of these genes between normal and epilepsy samples by the Wilcox test through the R package “limma”. The differentially expressed genes were identified by *p*-value < 0.05. The results were visualized as a box plot and heatmap using the R package “pheatmap” and “ggpubr”.

### 2.4. Selection of Disease-Related Genes

In order to select the disease-related genes that contribute the most to the onset of epilepsy from 21 genes, two common machine learning algorithms called random forests (RF) and a support vector machine (SVM) were constructed based on the expression data of 142 peripheral blood samples. Each model went through a 5-fold cross-validation using the “caret” package. “DALEX” package was used to analyze these two models. The residual distributions were visualized as boxplots and reverse cumulative distribution curves. According to the residuals, RF was eventually chosen to calculate the importance score of the genes. Since the number of classification trees needed to be set when building a random forest model, we compared the changes in the error rate corresponding to different numbers of trees and selected the number of trees corresponding to the minimum error rate to build the model. In the RF model, the classification became more stable as the Gini index decreases. Five adipocytokine pathway-related genes (RELA, PRKAB1, TNFRSF1A, CAMKK2, and CPT1B) were screened out based on the “Mean Decrease Gini”.

### 2.5. Construction and Evaluation of a Nomogram

The nomogram was constructed using the “rms” package in R to predict the risk of epilepsy. Based on the multivariate logistic regression, the regression coefficient of each gene, namely the coef value, was obtained. By standardizing the coef values, contributions of these genes to epilepsy were scored and visualized. The disease prediction ability of the nomogram was assessed using the “rmda” package. The calibration curve was used to compare the nomogram-predicted probability and actual probability. The decision curve analysis (DCA) and clinical impact curve were also conducted to evaluate the predictive accuracy of the nomogram.

### 2.6. Consensus Clustering of 21 Adipocytokine Pathway-Related Genes

Based on the expression profiles of 21 adipocytokine pathway-related genes, a cluster analysis was conducted using the “ConsensusClusterPlus” R package. The optimal number of subtypes was determined according to the cumulative distribution function curves. The result of classification was further verified through a principal component analysis (PCA) and was visualized by the “ggplot2” package.

### 2.7. Identification of Immune Microenvironment

A single-sample gene-set enrichment analysis (ssGSEA) algorithm was used by the “GSVA” R package to calculate enrichment scores of infiltrating immune cells in epilepsy. The summarized gene set of 23 immune cells shown in Appendix A was obtained from a previous study by Charoentong et al. [15]. Furthermore, the infiltration results were visualized as a boxplot using the “ggpubr” R package. Meanwhile, the correlation between the expression of these 21 genes and 23 infiltrating immune cells was explored.

### 2.8. Functional Enrichment Analysis of Distinct Adipocytokine Signal Patterns

The empirical Bayesian approach of the “limma” R package was used to find out differentially expressed genes (DEGs) between cluster A and B. The filter criteria were set as |logFC| > 0.5 and adjusted *p*-value < 0.05. Basing on these DEGs, Gene Ontology (GO) and KEGG pathway analyses were conducted to investigate its biological functions. The filter criterion was set as q-value < 0.05 using the “clusterProfiler” R package. The flowchart of our study is shown in Figure 1.

## 3. Results

### 3.1. The Landscape of Adipocytokine Pathway-Related Genes in Normal and Epilepsy Samples

To comprehensively investigate the potential pathogenesis of epilepsy, gene expression profiles were used to conduct Gene Set Enrichment Analysis (GSEA). As a powerful analytical method for interpreting data at the gene-set level, GSEA has long been used for biological function assessments. In this study, the dataset GSE143272 was used, and 142 peripheral blood samples were collected, including 51 normal samples and 91 epilepsy samples (idiopathic, n = 42; cryptogenic, n = 36; symptomatic, n = 13) [16,17]. Enrichment results revealed that the adipocytokine signaling pathway enriched apparently (normalized enrichment score = 1.689, nominal *p*-value = 0.010) in the epilepsy group (Figure 2a). We received the list of adipocytokine pathway-related genes from the KEGG pathway database (pathway: hsa04920). Among them, 43 genes were detected in the peripheral blood samples. Differential gene-expression analysis further identified 21 genes with statistical significance, consisting of 18 up-regulated genes (ACSL4, ADIPOR1, ADIPOR2, AKT1, CAMKK2, CPT1A, CPT1B, JAK2, NFKB1, NFKBIE, PCK2, PRKAB1, PRKAG1, RELA, RXRA, STAT3, TNFRSF1A, and TNFRSF1B) and 3 down-regulated genes (LEP, PRKAB2, and PRKCQ) in epilepsy cases (Figure 2b,c). These results indicated that the expression patterns of these genes are highly heterogeneous between normal and epilepsy samples.

### 3.2. Identification of a Five-Gene Signature for Epilepsy

To better understand what role adipocytokine signaling pathways play in epilepsy, a series of machine learning and bioinformatics methods were used. Expression profiles of 142 samples were applied to two different machine learning algorithms, named random forest (RF) and support vector machine (SVM). We visualized the residuals in two models using R package ggplot2 (Figure 3a,b) and compared them to obtain a more suitable model. The RF turned out to have smaller residuals and was chosen for subsequent analysis. We next performed an examination of the relationship between the error rate and the number of classification trees (Figure 3c) and built an RF model for signature screening. As shown here (Figure 3d), 21 adipocytokine pathway-related genes were arranged in sequence according to the value of the “Mean Decrease Gini”. Since the stability of the RF model increases as the Gini index decreases, the five most pivotal genes (RELA, PRKAB1, TNFRSF1A, CAMKK2, and CPT1B) were selected as a diagnostic signature to plot a nomogram. Furthermore, we found that antiepileptic drug treatment did not affect the expression of these five genes (Appendix A).

### 3.3. Construction and Evaluation of a Diagnostic Nomogram

Based on the five-gene signature, a nomogram was established to calculate and predict the likelihood of epilepsy (Figure 4a). In this multivariate logistic regression model, the value of each variable was assigned a score, respectively, through the point-scale axis. By adding up the scores of all the variables, the final score was calculated to estimate the risk of epilepsy. Assessments of the nomogram’s predictive power were subsequently carried out. The calibration plot observed a small variation between the nomogram prediction and the actual circumstance (Figure 4b). The clinical benefits of nomograms were presented by decision curve analysis (DCA). As shown by the DCA curves, the net benefit received from the nomogram was higher than the treat-all and treat-none schemes when the threshold probability ranged from 0.1 to 1.0 (Figure 4c). We then drew a clinical impact curve (Figure 4d), which also confirmed that the nomogram was a reliable and accurate model for disease prediction. The above altogether support the adipocytokine signaling pathway’s crucial role in epilepsy.

### 3.4. Expression Patterns of Adipocytokine Pathway-Related Genes in Epilepsy

According to the expression profiles of these 21 genes, an unsupervised consensus clustering analysis was performed to classify 91 epilepsy samples into several clusters. It achieved optimal clustering stability when the clustering index was at k = 2 (Figure 5a–c). Hence, these samples could be split into two distinct clusters: 58 samples in cluster A and 33 samples in cluster B. A principal component analysis (PCA) was used to verify the classification, showing a remarkable difference between the two clusters (Figure 5d). A boxplot and heatmap were further plotted to have an intuitive knowledge of the gene expression patterns of clusters A and B (Figure 6a,b). As pictured here, 13 of the 21 genes were screened out as statistically significant. The expression levels of ACSL4, ADIPOR1, CAMKK2, CPT1A, CPT1B, JAK2, NFKBIE, RELA, STAT3, TNFRSF1A, and TNFRSF1B in cluster B were higher than that in cluster A, while the levels of PRKAB2 and PRKCQ were significantly decreased in cluster B, revealing the diversity of adipocytokine signaling expression patterns in epilepsy.

### 3.5. Immune Landscapes in Distinct Adipocytokine Signal Expression Patterns

To estimate the immune cell infiltration status among these distinct adipocytokine signal patterns, “GSVA” R package was used to calculate the enrichment score of each infiltrating immune cell. As illustrated in Figure 7a, the levels of immune cell infiltration diverged significantly in different epilepsy clusters. We found that five types of lymphocytes, namely, activated CD4 T cells, activated CD8 T cells, CD56bright natural killer cells, CD56dim natural killer cells, and natural killer T cells, were significantly enriched in cluster A. In comparison, the infiltration levels of activated dendritic cells, myeloid-derived suppressor cells, macrophages, monocytes, natural killer cells, neutrophils, plasmacytoid dendritic cells, type 2 T helper cells, and type 17 T helper cells were higher in cluster B. We then performed a correlation analysis to observe the relationship between these 21 genes and infiltrating immunocytes (Figure 7b). As detailed here, the activated CD8 T cells had the strongest positive correlation with PRKCQ and a negative correlation with JAK2, which indicates that expression levels of PRKCQ and JAK2 in epilepsy play an important role in activated CD8 T cell infiltration.

### 3.6. Biological Characteristics of DEGs in Different Patterns

To quantify the biological variations within two adipocytokine signal expression patterns, we obtained 239 differentially expressed genes (DEGs) (Appendix A) to perform a Gene Ontology (GO) biological process and KEGG functional enrichment analysis. The top six most significant GO terms for biological process (BP), cellular component (CC), and molecular function (MF) were presented here to make an overall observation of the biological characteristics (Figure 8). Detailed comments on the GO enrichment results were shown in Table 1. Through GO analysis of the biological process, we found that DEGs were significantly enriched in the activation of immune response, regulation of innate immune response, lymphocyte differentiation, positive regulation of cytokine production, and so forth. For the cellular component subgroup, the DEGs were primarily enriched in the external side of the plasma membrane, secretory granule membrane, and so forth. In the molecular function subgroup, the DEGs were enriched in immune receptor activity, NAD+ nucleosidase activity and IgG binding (Figure 9a). Moreover, KEGG enrichment analysis revealed that DEGs mainly take part in the hematopoietic cell lineage, PD-L1 expression, and PD-1 checkpoint pathway in cancer and viral infections, such as Epstein-Barr virus infection and Measles (Figure 9b).

## 4. Discussion

As a common chronic neurological disorder, epilepsy accounts for a significant proportion of the world’s disease burden, causing the loss of approximately 13 million disability-adjusted life years (DALYs) and 60,000 premature deaths each year [18]. In spite of the considerable research interpreting the immune system’s role in epilepsy through different perspectives [19], more studies are needed to fully understand how epilepsy arises and progresses. Meanwhile, several studies on adipocytokines have sought to unveil its relevance to epilepsy [20,21].

In this study, GSEA was conducted to give a general description on the biological characteristics of epilepsy. Using a random forest algorithm, the impact of adipocytokine pathway-related genes was explored and a five-gene signature was selected to construct a diagnostic model. Epilepsy samples were gathered into several subtypes with huge variations in gene expression patterns, and their differences in immune cell infiltration were calculated. Differentially expressed genes (DEGs) were identified afterward, followed by a GO and KEGG functional analysis, to fully understand the impact of these genes on the epileptic pathologic process.

An abnormal activation of the adipocytokine pathway in peripheral blood of epilepsy samples was estimated. Adiponectin is an insulin-sensitizing adipocytokine with extensive anti-inflammatory, anti-atherogenic, and anti-apoptotic properties [22]. We found that ADIPOR1 and ADIPOR2, as cognate receptors of adiponectin, were highly expressed in the epilepsy group. CPT1A and CPT1B, as key rate-limiting enzymes in the mitochondrial fatty acid oxidation (FAO) pathway [23], also showed an upward trend compared with the control group. In contrast, the level of LEP, another well-known adipocytokine mentioned above, showed a downward trend in epilepsy. A similar phenomenon had been observed in clinical trials by other researchers [11]. Subsequent results of GSEA proved to be consistent with our findings.

Through machine learning algorithms, five adipocytokine pathway-related genes (RELA, PRKAB1, TNFRSF1A, CAMKK2, and CPT1B) robustly correlated with epilepsy were screened out as molecular biomarkers for disease prediction and could assist neurologists in making more accurate diagnoses, not only using the patient’s symptoms, MR imaging, EEG, etc. RELA, as a member of the NF-κB/Rel family of transcription factors, participates in diverse cellular responses like immunity, inflammation, tumorigenesis, apoptosis, and differentiation in the form of homodimers or heterodimers [24]. Wang et al. reported that RELA exhibits certain promoting effects on atherosclerosis-associated human aortic vascular smooth muscle cell proliferation and migration [25]. In tendon adhesion, RELA promoted a fibrous tissue formation and inflammatory reaction, yet suppressed apoptosis [26]. Our results showed that the expression level of RELA was higher in the epilepsy group. PRKAB1, a regulatory subunit of AMPK, was reported to promote fatty acid oxidation by phosphorylating acetyl-CoA carboxylase [27] and restrained hepatic lipogenesis via the miR-802/AMPK axis [28]. Another research study highlighted the impacts of glucagon in energy metabolism by activating the CAMKK2/AMPK pathway [29]. Upregulation of PRKAB1 and CAMKK2 in this study underscored the pivotal role of energy homeostasis in epilepsy. TNFRSF1A was defined as a susceptibility gene in tumor necrosis factor receptor-associated periodic syndrome (TRAPS) and multiple sclerosis (MS) [30,31], which also plays certain roles in apoptosis and inflammation. The activation of CPT1B, mentioned above, was found indispensable to the FAO process in skeletal muscle [23]. CPT1B-knockout mice were previously reported to have suppressed inflammatory responses [32]. The levels of these five genes all showed an upward trend in the epilepsy samples, suggesting a link between inflammation and epilepsy. The nomogram, using RELA, PRKAB1, TNFRSF1A, CAMKK2, and CPT1B, was next confirmed as a reliable predictive model for epilepsy that would bring patients a net clinical benefit when the threshold probability ranged from 0.1 to 1.0. Based on these results, the prominence of the adipocytokine pathway in epilepsy was further validated.

Consensus clustering verified the existence of distinct gene expression patterns in epilepsy. Cluster B had higher expression levels of ACSL4, ADIPOR1, CAMKK2, CPT1A, CPT1B, JAK2, NFKBIE, RELA, STAT3, TNFRSF1A and TNFRSF1B and lower levels of PRKAB2 and PRKCQ than that in cluster A. The same trend could be found between the epilepsy group and control group, indicating a more pronounced abnormal activation of the adipocytokine pathway in cluster B. As prior research discussed [33], adipocytokines were closely connected to many aspects of immunity and inflammation. Consequently, we performed single-sample gene-set enrichment analysis (ssGSEA) to calculate infiltrating immune cells. Subsequent results illustrated that cluster B was tremendously rich in innate immune cell infiltration, including neutrophils, natural killer cells, macrophages, monocytes, activated dendritic cells, plasmacytoid dendritic cells, and myeloid-derived suppressor cells, while several adaptive immune cells, called type 2 T helper cells and type 17 T helper cells, were also upregulated. Accumulating evidence has emphasized that peripheral innate immune cells that infiltrate into the central nervous system are strongly correlated with the progression of epilepsy [34,35]. Cluster A, in contrast, was characterized by the activation of lymphocytes including activated CD4 T cells, activated CD8 T cells, CD56bright natural killer cells, CD56dim natural killer cells, and natural killer T cells. Likewise, Alvarado et al. and Helmstaedter et al. also revealed a link between lymphocyte activity and limbic encephalitis [36,37]. Both innate and adaptive immunity play an important role in the pathogenesis of epilepsy [38]. In the brain tissue of epileptic patients, innate immune cells, such as microglia and astrocytes, are significantly activated. With the destruction of the blood-brain barrier, the peripheral immune system also responds to the abnormally activated innate immunity in the brain [19]. In the present study, the significant activation of innate immune cells, such as neutrophils, natural killer cells, and macrophages in cluster B, reflected a stronger nonspecific host response in these samples. These responses produced more cytokines and inflammatory mediators which contribute to the progression of epilepsy. Activation of the adaptive immune system, specifically T and B lymphocytes, is thought to be associated with viral infections and autoimmune diseases, which may lead to neuronal loss [39]. Thus, the innate and adaptive immune responses were more pronounced in the second group of epilepsy samples with a significant activation of the adipocytokine signaling pathway, which may contribute to the persistence of epilepsy and the progression of clinical symptoms. Following GO enrichment analyses showed that DEGs between clusters A and B were significantly enhanced for innate and adaptive immunity. Meanwhile, DEGs in KEGG were mainly linked to hematopoietic cell lineage, suggesting an intimate connection with blood cell differentiation.

However, our research has certain limitations. First, due to the small number of publicly available microarray datasets, we could not validate the results with another cohort. Second, the results of gene expression and immune cell infiltration have yet to be proved by actual experiments. Third, the variations in clinical symptoms and prognosis caused by different gene expression patterns need to be further determined on account of the absence of clinical characteristics. Therefore, large scale studies with adequate clinical information are urgently needed to strengthen the recognition of the pathological mechanisms of epilepsy.

## 5. Conclusions

In conclusion, the adipocytokine signaling pathway is an active participant in the pathogenesis of epilepsy. Its predictive capability for epilepsy and impacts on the peripheral immune landscape has been validated, and thus may provide a novel insight into the underlying mechanism of epilepsy.

## Figures and Tables

**Figure 1 brainsci-12-01156-f001:**
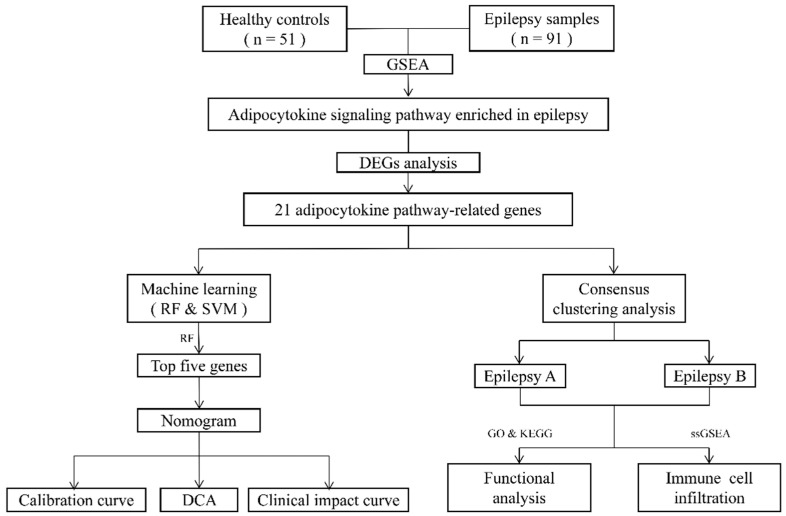
The flowchart of this study.

**Figure 2 brainsci-12-01156-f002:**
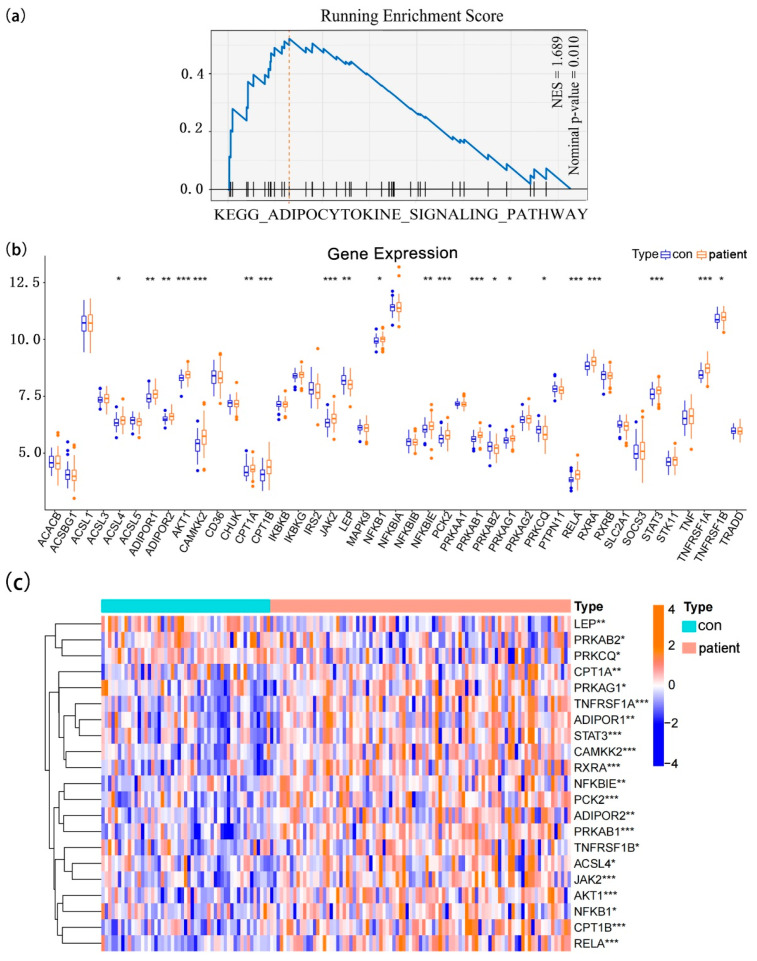
Peripheral blood expression landscapes of adipocytokine pathway-related genes and GSEA in patients with epilepsy. (**a**) GSEA shows that the adipocytokine signaling pathway is significantly enriched in epilepsy samples. (**b**) The expression levels of 43 genes in the normal and epilepsy samples and 21 genes were statistically significant. (**c**) The heatmap contains 21 differential expressed genes, of which 3 were down-regulated and 18 were up-regulated in epilepsy samples. NES, normalized enrichment score. * *p* < 0.05, ** *p* < 0.01, *** *p* < 0.001.

**Figure 3 brainsci-12-01156-f003:**
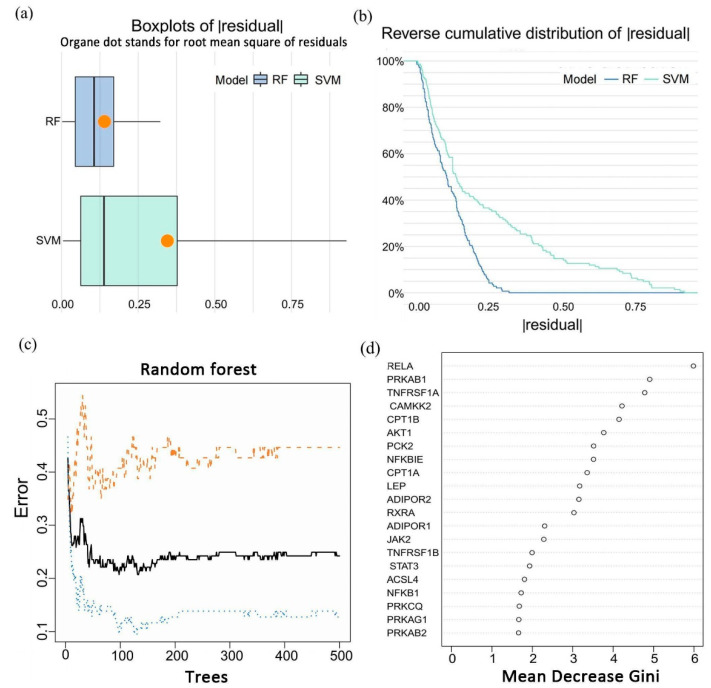
Identification of a five-gene signature in epilepsy. (**a****,b**) Residual comparison between RF and SVM models shows that RF has a smaller residual. (**a**) The comparison of residuals between RF and SVM models shows smaller residuals in the RF model. RF, random forest; SVM, support vector machine. (**b**) The comparison of residuals between RF and SVM models shows smaller residuals in the RF model. RF, random forest; SVM, support vector machine. (**c**) The error rate varies with the number of classification trees in the random forest model. Setting trees with the appropriate number will reduce the error rate of the RF model. The blue line represents the error rate in the healthy controls, the yellow line for epilepsy samples, and the black line for all samples. (**d**) The importance scores of 21 candidate genes are sorted based on the Gini index. The lower the Gini index, the more accurate the classification results. The top 5 genes were chosen for subsequent analysis.

**Figure 4 brainsci-12-01156-f004:**
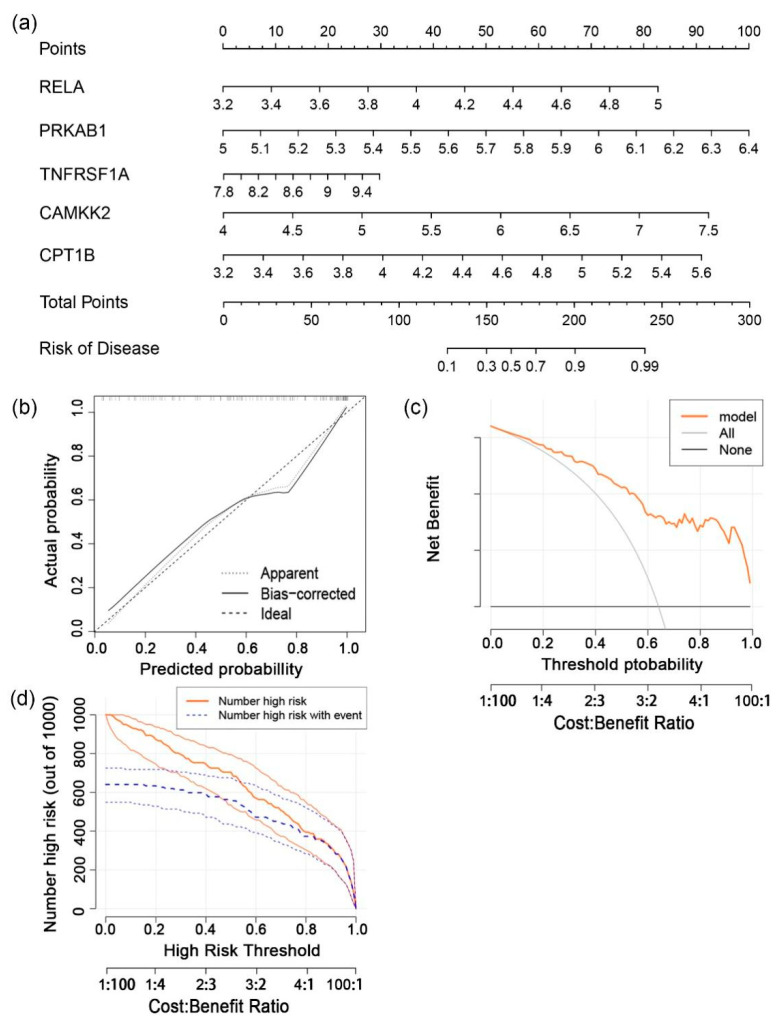
Construction and evaluation of a nomogram. (**a**) A nomogram that calculates the risk of epilepsy based on 5 select predictive genes using a multivariate logistic regression model. (**b**) The calibration curve shows a small difference between the predicted probability and the actual probability. (**c**) Decision curve analysis shows that the nomogram is clinically useful when the threshold ranges from 0.1 to 1.0. (**d**) The clinical impact curves show that the nomogram becomes more accurate as the threshold increases. The orange curve represents samples classified as epilepsy and the blue curve for actual epilepsy samples.

**Figure 5 brainsci-12-01156-f005:**
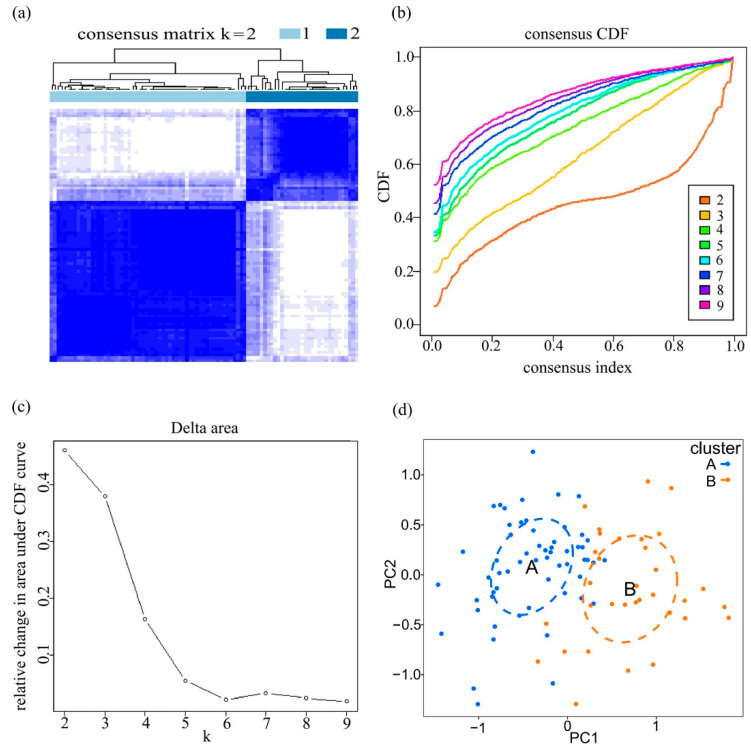
Consensus clustering of 21 adipocytokine pathway-related genes. (**a**) 91 epilepsy samples were grouped into two clusters according to the consensus clustering matrix (k = 2). (**b**) Cumulative distribution function (CDF) curve when k ranges from 2 to 9. (**c**) Changes in area under CDF curve when k ranges from 2 to 9. (**d**) Principal component analysis shows a good differentiation between the two clusters. The blue and orange dots represent clusters A and B, respectively.

**Figure 6 brainsci-12-01156-f006:**
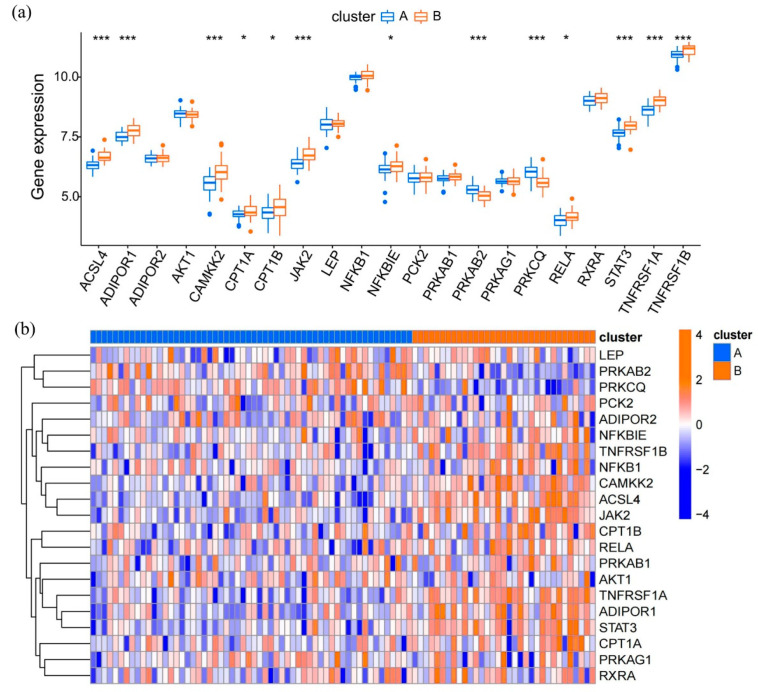
Expression of adipocytokine signaling pathway among different expression patterns in epilepsy samples. (**a,b**) A boxplot and heatmap show the expression levels of these 21 genes in cluster A and B. * *p* < 0.05, *** *p* < 0.001.

**Figure 7 brainsci-12-01156-f007:**
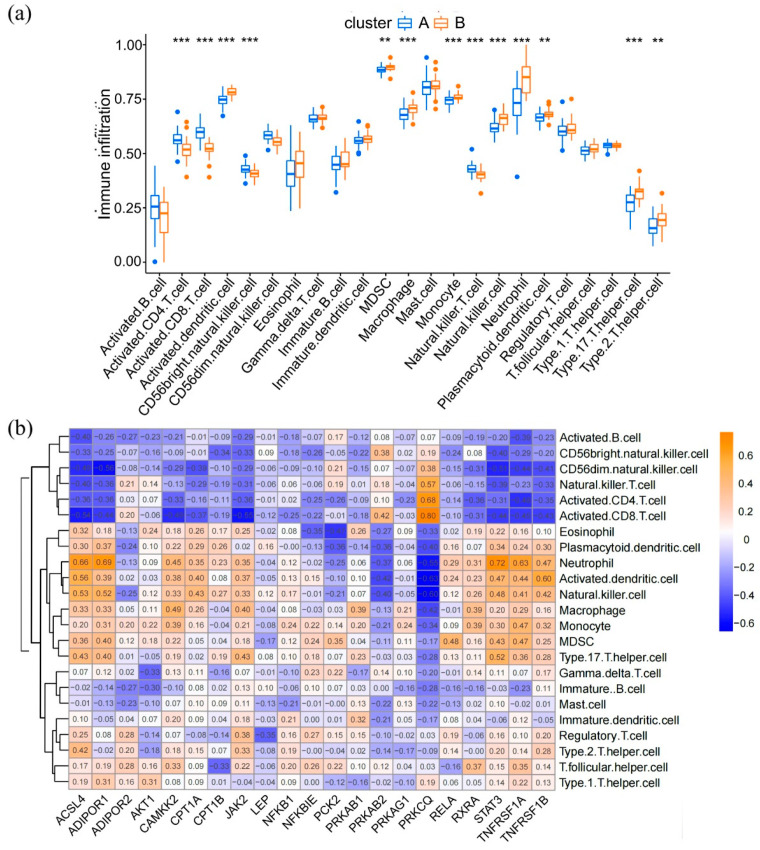
Characteristics of immune cell infiltration in different gene expression patterns and the correlation of immune cells with individual genes. (**a**) Abundance of 23 infiltrating immune cells in different clusters. (**b**) The correlation between the infiltrating immune cells and the adipocytokine pathway-related genes. ** *p* < 0.01, *** *p* < 0.001.

**Figure 8 brainsci-12-01156-f008:**
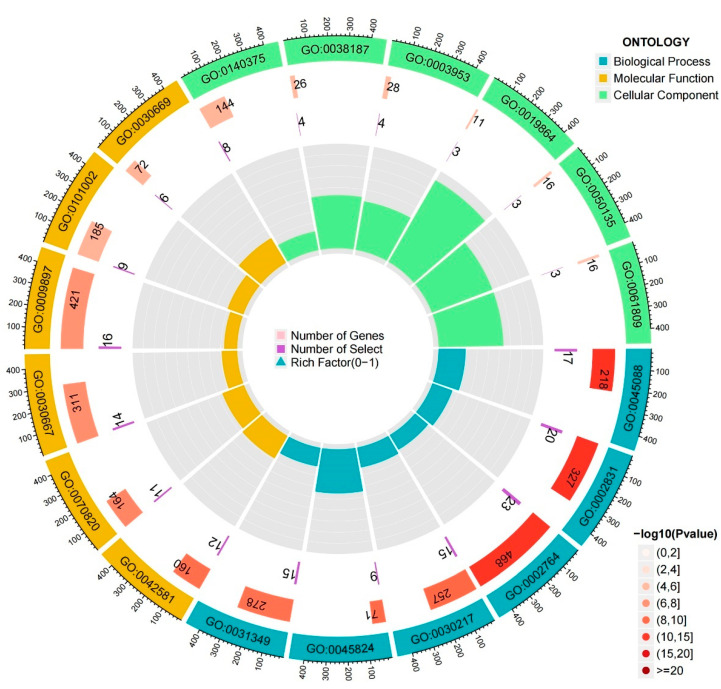
Distribution of DEGs in GO enrichment analysis. The order of the circles is from the outside in. The first circle shows the most significantly enriched terms of biological process (BP), cellular component (CC), and molecular function (MF), with the scale outside corresponding to the number of genes. The second circle shows the number of background genes in the corresponding GO terms and the *p*-value. The third circle shows the number of DEGs. The inner circle shows the enrichment score of the corresponding GO terms.

**Figure 9 brainsci-12-01156-f009:**
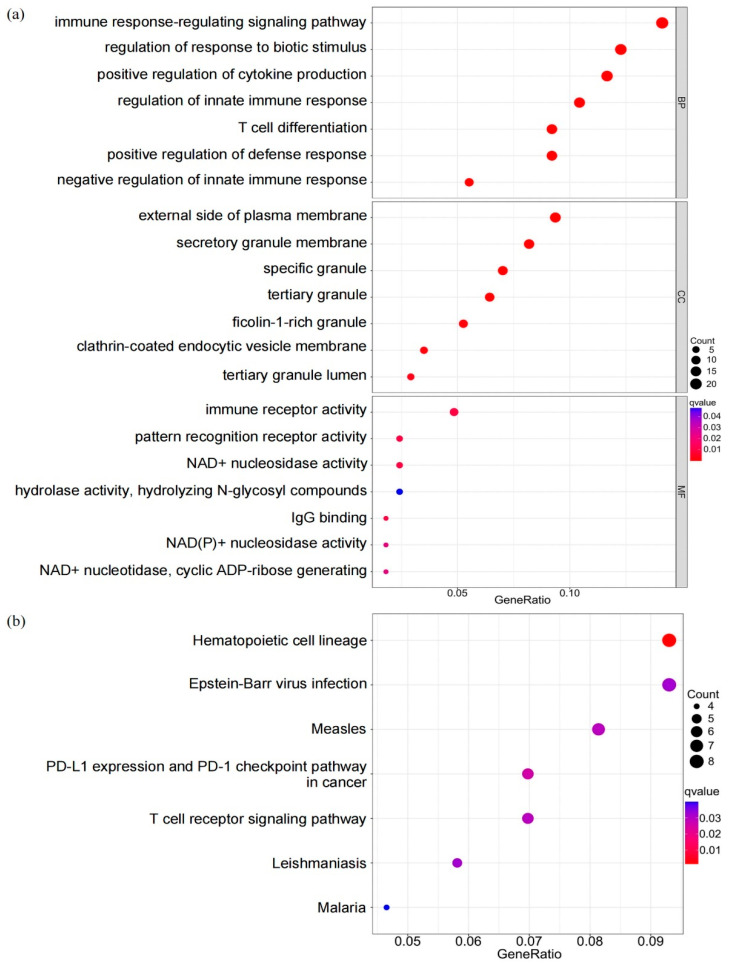
Biological function analysis based on the DEGs. (**a**) Bubble plot of GO enrichment analysis. (**b**) Bubble plot of KEGG enrichment analysis. The *x*-axis represents the ratio of pathway-enriched DEGs to the total DEGs. The size and color of bubbles correspond to number of genes and q-value of the enrichment significance, respectively.

**Table 1 brainsci-12-01156-t001:** The top six most significant GO terms enriched by the 239 DEGs.

**ID**	**Categories**	**Description**
GO:0140375	CC	immune receptor activity
GO:0038187	CC	pattern recognition receptor activity
GO:0003953	CC	NAD+ nucleosidase activity
GO:0019864	CC	IgG binding
GO:0050135	CC	NAD(P)+ nucleosidase activity
GO:0061809	CC	NAD+ nucleotidase, cyclic ADP-ribose generating
GO:0045088	BP	regulation of innate immune response
GO:0002831	BP	regulation of response to biotic stimulus
GO:0002764	BP	immune response-regulating signaling pathway
GO:0030217	BP	T cell differentiation
GO:0045824	BP	negative regulation of innate immune response
GO:0031349	BP	positive regulation of defense response
GO:0042581	MF	specific granule
GO:0070820	MF	tertiary granule
GO:0030667	MF	secretory granule membrane
GO:0009897	MF	external side of plasma membrane
GO:0101002	MF	ficolin-1-rich granule
GO:0030669	MF	clathrin-coated endocytic vesicle membrane

GO, gene ontology; DEGs, differentially expressed genes; CC, cellular component; BP, biological process; MF, molecular function; NAD+, nicotinamide adenine dinucleotide; ADP, adenosine diphosphate.

## Data Availability

All data in this study are available in the article/Appendix A.

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
