# Peer review of "Identification of Adipocytokine Pathway-Related Genes in Epilepsy and Its Effect on the Peripheral Immune Landscape"

_brainsci, 2022, doi:10.3390/brainsci12091156_

Round 1
Reviewer 1 Report
this research elucidates the predictive value of adipocytokines in epilepsy. better to classify the result regarding the ethiology of epilepsy
Author Response
Thank you for your comments. We have revised the article.
Reviewer 2 Report
The paper deals with an interesting analysis to detect genetic expression of adipocytokines in patients with epilepsy, based on previous work, to eventually apply it to the clinic and use it as precision medicine for the treatment of this group of patients.
The most interesting thing about the work is that they find differential expression in 21 genes and that they can classify the cases into two different types of expression. However, they should validate it in another population, as mentioned in the limitations. For the validity of the findings, it is essential to correlate them with some type of clinical expression of epilepsy. Therefore, I list below the most relevant points to clarify in the text, before publication. 1) They do not have clinical or other cohort clinical data to validate results 2) The number of samples is not that large, so the scope of these results is very limited. 3) They should better explain clinical characteristics of the patients selected for analysis: are they refractory? 4) Is it possible that the changes in the expression of these genes are due to the medication they receive? 5) Regarding the nomogram: What would be the values ​​to use? Do they correspond to fold-change in the expression? How would they be determined? 6) These values ​​do not change regardless of the methodology used to estimate gene expression? 7) How do you propose the translation to clinical practice? Minor comments: - They use of the acronym RF (random forest) without defining materials and methods. They define it in results (later) and it should be the other way around.- Kyoto Encyclopedia of Genes and Genomes (KEGG) was defined more than once.
Author Response
Thank you for your comments. We have revised the article and attached the reply information at the end of the article. Please see the attachment.

Reviewer 3 Report
In their present work, Chen et al. describe a complex approach to identify genes of the adipocytokine pathway. Furthermore, the effects of these genes on the peripheral immune landscape will be investigated and illustrated.
Some points to improve:
- The concept, which is in itself accurately planned and involves the use of many methods, remains somewhat confusing and difficult for the reader to understand. An additional graphic, which gives the overall context of the individual experimental/methodological steps, would be desirable and helpful here.
- The title promises the presentation of the effects on the peripheral immune landscape, which in my opinion is done somewhat diffusely at the end. The reader is left alone in the discussion with the interpretation and classification of the diffusely presented data over large parts. The consequence of the results is presented too little pointedly and therefore remains unclear. A large number of data are mentioned, but are strung together incoherently. Thereby, the many graphs are good in presentation, but are poorly embedded and explained.
- Figure colors: Green and red should also be avoided in the color choice. Please change.
- The figure legends are kept short, which in my opinion makes them difficult to understand.
- The English language is basically good, but could be improved in some places (e.g. line 40 "It is characteristic by ..." and line 102 "...was used to analyze...").
Author Response
Thank you for your comments. We have revised the article and attached the reply.
Response to Reviewer 3 Comments
Point 1:The concept, which is in itself accurately planned and involves the use of many methods, remains somewhat confusing and difficult for the reader to understand. An additional graphic, which gives the overall context of the individual experimental/methodological steps, would be desirable and helpful here.
Response 1: Thank you very much for your suggestion. We have added a flow chart (Figure 1) in an updated version.
Point 2:The title promises the presentation of the effects on the peripheral immune landscape, which in my opinion is done somewhat diffusely at the end. The reader is left alone in the discussion with the interpretation and classification of the diffusely presented data over large parts. The consequence of the results is presented too little pointedly and therefore remains unclear. A large number of data are mentioned, but are strung together incoherently. Thereby, the many gra phs are good in presentation, but are poorly embedded and explained.
Response 2: Thanks for your suggestion. We have modified the discussion section to focus on the impact on immune landscape (line 386).
Point 3:Figure colors: Green and red should also be avoided in the color choice. Please change.
Response 3: Thanks for your suggestion. We have changed the green and red to blue and orange in all figures.
Point 4:The figure legends are kept short, which in my opinion makes them difficult to understand.
Response 4: Thanks for your suggestion. We have rewritten the figure legends for clarity.
Point 5:The English language is basically good, but could be improved in some places (e.g. line 40 "It is characteristic by ..." and line 102 "...was used to analyze...")
Response 5: Thank you for pointing out our mistakes. We have rewritten these sentences (line 40 and line 102) and rechecked the full text to correct the inappropriate words.
Reviewer 4 Report
Dear author,
I hope my report finds you well and safe. The aim of the current study was to discover the genetic association of the role of adipocytokines in antiepileptic and modulating immune responses in the pathological process of epilepsy and its impacts on peripheral immune characteristics. That sounds interesting. However, the feature of the chosen sample is mandatory to be stated, including the age, sex, race, diseases, weight, and nature of their diet, to provide a clear outlook of the obtained conclusion.
Yours
Author Response
Thank you for your comments. We have revised the article and attached the reply.
Response to Reviewer 4 Comments
Point 1:However, the feature of the chosen sample is mandatory to be stated, including the age, sex, race, diseases, weight, and nature of their diet, to provide a clear outlook of the obtained conclusion.
Response 1: Thanks for your suggestion. The reason we did not introduce detailed sample information, such as age, gender, antiepileptic drug etc., is that we analyze the data for gene set GSE143272. And the owner of GSE143272. published two papers with a table of detailed sample information.
In order to avoid duplication of tables, we cited those two articles in our paper as ref 16 and ref17.
- Rawat C, Kushwaha S, Srivastava AK, Kukreti R. Peripheral blood gene expression signatures associated with epilepsy and its etiologic classification. Genomics 2020 Jan;112(1):218-224. PMID: 30826443
- Rawat C, Kutum R, Kukal S, Srivastava A et al. Downregulation of peripheral PTGS2/COX-2 in response to valproate treatment in patients with epilepsy. Sci Rep 2020 Feb 13;10(1):2546. PMID: 32054883
Round 2
Reviewer 4 Report
Dear author,
I hope this letter finds you healthy and secure. Thank you for performing the required changes.
Yours